# The Effect of Polyphenols on Hypercholesterolemia through Inhibiting the Transport and Expression of Niemann–Pick C1-Like 1

**DOI:** 10.3390/ijms20194939

**Published:** 2019-10-06

**Authors:** Shoko Kobayashi

**Affiliations:** Research Center for Food Safety, Graduate School of Agricultural and Life Sciences, The University of Tokyo, 1-1-1 Yayoi, Bunkyo-ku, Tokyo 113-8657, Japan; ashoko@mail.ecc.u-tokyo.ac.jp; Tel.: +81-3-5841-5378

**Keywords:** flavonoid, Niemann–Pick C1-like 1, luteolin

## Abstract

The Niemann–Pick C1-like 1 (NPC1L1) protein is a cholesterol transporter that is expressed in the small intestine. This report describes the discovery of NPC1L1, its transport properties, and the inhibitory effects of polyphenols on NPC1L1. NPC1L1 was identified in 2004 while searching for ezetimibe molecular targets. Excessive synthesis of cholesterol results in hyperlipidemia, which increases the amount of bile cholesterol excreted into the duodenum. The inhibition of NPC1L1 decreases blood cholesterol because food and bile cholesterol are also absorbed from NPC1L1 in the intestine. Some polyphenols, particularly luteolin, have been reported as NPC1L1-mediated anti-dyslipidemia constituents. Luteolin affects NPC1L1 through two mechanisms. Luteolin directly inhibits NPC1L1 by binding to it, which occurs in a short timeframe similar to that for ezetimibe. The other mechanism is the inhibition of NPC1L1 expression. Luteolin reduced the binding of Sterol-regulatory element-binding protein 2 (SREBP2) in the promoter region of the *NPC1L1* gene and decreased mRNA levels of *SREBP2* and *hepatocyte nuclear factor 4α*. These data suggest that luteolin decreases the expression of NPC1L1 through regulation of transcription factors. This review also explores the effect of other polyphenols on NPC1L1 and hypercholesterolemia.

## 1. Identification of Cholesterol Transporter

Cholesterol is an essential component of every cell in the body, endowing cell membranes with strength and flexibility. It is also essential for many of the body’s metabolic processes, including the production of steroid hormones, bile, and vitamin D. However, too much cholesterol in the blood can increase the risk of serious blood-vessel-related diseases, such as cardiac infarction and cerebral apoplexy. [1]. Statins are regarded as a first-line therapy, and they inhibit 3-hydroxy-3-methyl-glutaryl-CoA reductase, a component of the mevalonate pathway that affects cholesterol biosynthesis, to reduce the de novo synthesis of cholesterol [2]. However, even with the maximal tolerated statin dose, it is difficult to achieve the target level of low-density lipoprotein cholesterol (LDL-C) in some patients, set according to their risk profile. Such findings suggest that the needs of patients on statin treatment are not met. Through pharmaceutical drug screening, a new drug with a separate mechanism to statins, called ezetimibe, was found to be potentially effective in treating hypercholesterolemia [3]. During chronic preclinical studies, it was determined that ezetimibe inhibited the rise in plasma cholesterol normally observed with cholesterol feeding in rhesus monkeys [4]. Although the mechanism behind the activity of ezetimibe was unknown, a clinical study of it was initiated. Because cholesterol is hydrophobic, it has been thought to be transported across the brush border membrane through passive diffusion. However, the discovery of ezetimibe contributed to the discovery of the cholesterol transporter Niemann–Pick C1-like 1 (NPC1L1). NPC1L1 was identified through a search for ezetimibe molecular targets. [5]. Altmann et al. used a genomic-bioinformatic screening approach to identify the genes involved in cholesterol uptake. They prepared cDNA libraries from rat jejunum mucosal scrapings and jejunum enterocytes [5]. The 16,500 expressed sequence tags (ESTs) derived from these libraries were combined with all available rat ESTs. The obtained rat ESTs were annotated by cross-referencing with both mouse and human sequence data. This sequence database was analyzed for all transcripts containing features anticipated to be present in a cholesterol transporter, such as sterol-sensing domain. Altmann et al. found one credible candidate gene from this analysis. It has approximately 50% amino acid homology to NPC1 [6], a gene the deficiency of which is associated with the cholesterol storage disease Niemann–Pick type C (NP-C) and that functions in intracellular cholesterol trafficking. They then created the NPC1L1 knockout mice. The KO mouse’s intestinal cholesterol absorption is as low as levels seen in ezetimibe-treated mice, and ezetimibe treatment results in no further reduction of intestinal cholesterol absorption in NPC1L1 knockout mice [5]. This study clearly demonstrated that NPC1L1 is the molecular target of ezetimibe, and NPC1L1 is a cholesterol transporter of the intestine.

When the NPC1L1 sterol-sensing domain (SSD) detects an increase in cholesterol in the microdomain, the NPC1L1 N-terminal binds a cholesterol molecule and an NPC1L1-complex is made; this complex internalizes through clathrin-mediated endocytosis [7,8,9]. Recently, Li et al. reported that a novel endocytic motif was identified in the cytoplasmic C-terminal tail of NPC1L1 (Figure 1) [10]. When cholesterol binds to the intercellular N-terminal loop of NPC1L1, the cytoplasmic C-terminal tail, which is called the YVNXXF motif, leaves the cell-membrane and binds to the Numb protein. Numb protein then binds to clathrin adaptor protein (AP) and clathrin. NPC1L1 is surrounded by the clathrin/AP2-coated vesicles and is transported into the cytoplasm by endocytosis. This complex is then transported to the recycling endosome. There the cholesterol separates from NPLC1L1 and moves to the endoplasmic reticulum. NPC1L1 returns to the cytoplasmic membrane to repeat the cycle. However, the reason why cholesterol transport is performed through endocytosis remains unclear, given its low energy efficiency.

NPC1L1 facilitates cholesterol transport through clathrin-dependent endocytosis. NPC1L1, which is surrounded by the clathrin/AP2-coated vesicles, is transported into the cytoplasm and eventually the recycling endosome. At this point, the cholesterol separates from NPLC1L1 and moves to the endoplasmic reticulum. Meanwhile, NPC1L1 returns to the cytoplasmic membrane. AP2 refers to adaptor protein 2.

Flotillins, which are proteins that make up lipid rafts and also mediate cholesterol endocytosis [11]. Ge et al. reported that ezetimibe binds NPC1L1 and blocks its cholesterol absorption and that ezetimibe also prevents the interaction of NPC1L1 with clathrin/AP2-coated vesicles [12]. Weinglass et al. found that the binding affinities of ezetimibe differed between dog and mouse by performing chimeric and mutational studies using the NPC1L1 mutant and a radioligand analog of ezetimibe [13]. The NPC1L1 found in dogs has a higher affinity for binding to ezetimibe than that in mouse. The affinity depends on the molecular determinants present in a 61-aa region of a large extracellular domain (loop C) where Phe-532 and Met-543 appear to be key contributors. Phe-532 and Met-543 are conserved in human NPC1L1.

## 2. Meaning of Inhibition of NPC1L1

Cholesterol in the intestine is derived from food (400–500 mg/day) and bile cholesterol (800–2000 mg/day) (Figure 2). Cholesterol is synthesized in the liver, after which it is delivered to the blood circulation and stored in the bladder and the peripheral tissues. Bile cholesterol is excreted into the duodenum after eating. Excessive cholesterol synthesis results in hyperlipidemia, which increases the amount of bile cholesterol excreted into the duodenum. The inhibition of NPC1L1 decreases blood cholesterol because food and bile cholesterol are also absorbed from NPC1L1.

Statins inhibit the synthesis of cholesterol in the liver, whereas ezetimibe inhibits the transport of cholesterol in the intestine. NPC1L1 transports dietary cholesterol and bile cholesterol. ASBT, apical sodium-dependent bile acid transporter; HDL, high density lipoprotein; LDL, low-density lipoprotein; NPC1L1, Niemann–Pick C1-Like 1.

## 3. NPC1L1 Transport Kinetic Parameters

Nekohashi et al. evaluated the cholesterol transport ability of Caco-2 cells, a human colon adenocarcinoma (Caco-2) cell line, because the NPC1L1 kinetic parameter had not been clearly [14]. They determined the levels of [1–^3^H(N)]-cholesterol and regular cholesterol transported into cells as a measure of radioactivity, using a liquid scintillation counter. The concentration-dependent uptake of cholesterol by Caco-2 cells obeyed a saturation process, indicating the occurrence of active and passive transport [14]. Kinetic parameters for NPC1L1 transport activity were obtained from the MULTY program [15]. The kinetic parameters of cholesterol uptake in Caco-2 cells *J*max, *K*t, and *K*d were 6.89 ± 2.96 µM, 19.03 ± 11.58 µM, and 0.11 ± 0.02 pmol/min/mg protein, respectively. The ezetimibe inhibition rate of cholesterol uptake in Caco-2 cells was 50%, which is almost equal to the data reported by Altman et al. [5].

## 4. NPC1L1-Mediated Anti-Dyslipidemia Effect of Polyphenols

Many types of polyphenols have been reported to have an effect on dyslipidemia. Table 1 summarizes the papers which have reported the effects of polyphenols and polyphenol-rich extracts on NPC1L1 both in vivo and in vitro.

There are seven polyphenols (Figure 3) that have been reported to inhibit NPC1L1; luteolin, curcumin, cyanidin-3-glucoside, chlorogenic acid, and catechin inhibit the expression of NPC1L1, whereas luteolin, quercetin, and epigallocatechin gallate (EGCG) inhibit the transport of NPC1L1.

There are two mechanisms through which polyphenols affect the activity of NPC1L1: inhibition of transport or expression. Using animal models, including Wister rats, Sprague–Dawley rats, C57bL/6 mice, and hamsters, it was demonstrated that polyphenols decreased the blood levels of cholesterol via NPC1L1 inhibition. The most reported polyphenol is luteolin, which has been shown to decrease *NPC1L1* mRNA levels in Caco-2 cells and mouse intestinal mucosa [16,17]. Curcumin, cyanidin 3-glucoside, catechin, and chlorogenic acid also suppress *NPC1L1* mRNA [18,19]. It has been reported that luteolin and epigallocatechin gallate (EGCG) inhibit NPC1L1 through direct binding [14,20]. The structure-activity relationship was not demonstrated. Feng et al. reported that hawk tea reduced blood cholesterol level. EGCG, which is a compound found in hawk tea, inhibits endocytosis of NPC1L1 from the plasma membrane to endocytic recycling compartment [20]. Additionally, luteolin and quercetin inhibit the transport of cholesterol by NPC1L1 through direct binding [14]; however, the mechanism through which this occurs is unclear. Luteolin is the only flavonoid that inhibits NPC1L1 through two distinct mechanisms, particularly by affecting its activity and expression.

Some reports suggest that flavonoid-rich extracts have an impact on NPC1L1. Most in vivo experiments utilize the extract because it is very expensive to use the pure compound for in vivo studies. Cranberry anthocyanin extract (CrA) contains peonidin 3-galactoside, peonidin 3-arabinoside, and cyanidin 3-galactoside and -arabinoside [21]. Plasma total cholesterol and aorta atherosclerotic plaque decreased in a dose-dependent manner with increasing amounts of CrA when added to a hamster diet [21]. However, CrA had no effect on the mRNA levels of intestinal *NPC1L1*. Blueberry anthocyanin extract mainly contains cyanidin 3-galactoside and petunidin 3-galactoside [22]. Incorporating blueberry anthocyanins into a diet down-regulated the expression of *NPC1L1*. These results contradict those seen with CrA, which is possibly due to use of an extract. It is unclear which compound in the cinnamon extract is involved in the downregulation of *NPC1L1* expression but the extract has been reported to reduce the intestinal *NPC1L1* mRNA level [23].

## 5. Luteolin Inhibit NPC1L1 Transport in the Caco-2 Cell Monolayer

Numerous epidemiological studies have reported a correlation between the consumption of polyphenols and a decreased risk of arteriosclerosis [24,25,26]. Nekohashi et al. hypothesized that polyphenols would affect the intestinal transport of cholesterol and screened 34 polyphenols in Caco-2 cells. They found that liquiritigenin, sakuranetin, isosakuranetin, hesperetin, apigenin, luteolin, quercetin, daidzein, coumestrol, phloretin, and (−)-EGCG significantly inhibited the uptake of cholesterol. Because luteolin and quercetin are present in herbs and edible plants, these flavonoids were selected for use in subsequent assays. Certain flavonoids, epigallocatechin gallate, and theaflavins inhibit the absorption of cholesterol by disrupting micelle formation [27,28]. Similarly, luteolin and quercetin inhibit micelle formation in the micellar inhibition assay; however, the luteolin and quercetin concentrations were so high in this assay that the inhibitory effect on the formation of micelles did not play any critical role in the uptake of cholesterol by Caco-2 cells.

There were two hypotheses remaining. The first was that these flavonoids inhibit cholesterol micelle formations, and the second that the flavonoids affect intestinal epithelial cells such as transporter. The inhibitory effect of these flavonoids on the formation of micelles did not play a critical role in the uptake of cholesterol by Caco-2 cells. So, Caco-2 was preincubated with flavonoids prior to the cholesterol uptake assay to determine whether the flavonoids affected intestinal epithelial cells. Following this, luteolin and quercetin were preincubated with Caco-2 cells, and the cholesterol uptake decreased in a dose-dependent manner (25–100 μM). These data suggested that luteolin and quercetin decreased the cholesterol absorption by affecting the Caco-2 cells.

There is a large amount of transporter expression in Caco-2 cells. Using an NPC1L1 expression vector transfected into HEK293T cells, which is null for transporter expression, to determine whether luteolin and quercetin influence NPC1L1. Cholesterol uptake increased in the NPC1L1 transfected group compared with that in the control mock-transfected group. On the other hand, ezetimibe, luteolin, and quercetin decreased cholesterol uptake at a concentration equal to that of the mock control group. These data suggested that these flavonoids affect NPC1L1 transport. Nekohashi et al. hypothesized two mechanisms behind the inhibitory effect of quercein and luteolin on NPC1L1 transport. One was that the NPC1L1 protein structure or the uptake route may have changed because cholesterol uptake by Caco-2 and NPC1L1 transfected cells was inhibited within 1 h of the addition of the flavonoids. Ezetimibe binding to the extracellular loop C of NPC1L1 causes some conformational changes in the NPC1L1 protein. This conformational change inhibits NPC1L1 and cholesterol interactions and cholesterol-induced NPC1L1 endocytosis [6,29]. One possibility is that these flavonoids also bind to some domain of NPC1L1 to result in conformational changes of this protein, thereby inhibiting NPC1L1. It was reported that NPC1L1 localizes to membrane microdomains (lipid rafts) enriched in cholesterol. The N-terminal is important for moving extracellular cholesterol to the membrane-localized SSD region, which creates a raft-like plasma membrane microdomain [8]. EGCG inhibits ileal bile acid transporter by suppressing plasma membrane cholesterol [30]. If flavonoids suppress cholesterol in the microdomain, its composition may change, resulting in indirect inhibition of NPC1L1 transport.

## 6. Luteolin and Quercetin Decreased the Blood Cholesterol Level in Rats

Nekohashi et al. also evaluated luteolin and quercetin activity in vivo [14]. Male Wistar rats were divided into four groups: a no cholesterol group (NC) fed with AIN-93G diet, a normal diet made in our laboratory from essential ingredients, and orally administered 1% DMSO; a high cholesterol group (HC) fed with 0.5% cholesterol added to AIN-93G diet and orally administered 1% DMSO; an HL group fed with 0.5% cholesterol added to AIN-93G diet and orally administered 20 mM luteolin diluted in 1% DMSO at 5 mL/kg body weight; and a HQ group fed with 0.5% cholesterol added to AIN-93G diet and orally administered 20 mM quercetin diluted in 1% DMSO at 5 mL/kg body weight. After the 9 days, the serum cholesterol concentrations increased in the HC group compared with the NC group. However, the serum cholesterol concentrations decreased in the HL and HQ groups compared with the HC group. These findings suggest that the inhibition of cholesterol absorption by luteolin and quercetin may also contribute to the decrease in blood cholesterol concentration in vivo.

Won et al. also reported that luteolin supplementation impedes the elevations of plasma lipid and cholesterol induced by high-fat consumption in mice [17]. They fed a high-fat diet and luteolin to mice. After 8 weeks of treatment, the administration of 250 ppm luteolin modulated the total and serum levels of non-HDL cholesterol. In the liver samples, decreases in TC and total TG were observed in mice fed 250 ppm luteolin. Mice receiving 250 ppm luteolin exhibited increased fecal cholesterol output.

## 7. The Effect of Luteolin on NPC1L1 Expression

In previous in vivo study [14], the serum cholesterol levels in the NC group remained constant throughout the 9 experimental days. The serum cholesterol levels of the HC group continued to increase during this experiment. However, increases in the cholesterol concentrations for the HL and HQ groups were inhibited: they were smaller than in the HC group and as small as the NC group. It was hypothesized that luteolin and quercetin inhibit the expression of NPC1L1 because of the long feeding time in vivo. Ogawa et al. checked if the *Npc1l1* expression of the HC group had a tendency to be higher than that of the NC group [16]. The *Npc1l1* expression of the HL group was the same as that of the NC and HC groups. The *Npc1l1* expression of the HQ group was lower than that of the HC group. These results suggested that the intake of cholesterol resulted in an increase of *Npc1l1* expression in rat intestine, whereas quercetin decreased the expression of *Npc1l1.* Treatment of Caco-2 cells with luteolin for 48 h significantly decreased levels of NPC1L1 mRNA and protein. Rat intestinal *NPC1L1* mRNA levels decreased in the HQ group; however, quercetin did not affect the protein levels in human intestinal epithelial cells Caco-2. These results suggest that the effects of quercetin and luteolin on NPC1L1 expression may be species-specific. After 48 h of incubation with luteolin in Caco-2 cells, cholesterol uptake was also decreased. Luteolin and quercetin both decreased cholesterol uptake when these flavones were preincubated with Caco-2 cells for 1 h [14]. These data suggest that luteolin may inhibit NPC1L1 transport by two mechanisms. One is the direct inhibition of NPC1L1 by binding to it, which occurs in a short timeframe like for ezetimibe [14]. The other mechanism is the inhibition of NPC1L1 expression by luteolin [16]. Won et al. reported that *Ncp1l1* mRNA was downregulated and *Abcg-5/8* mRNA, which encodes a cholesterol efflux transporter, was upregulated in the intestinal mucosa of luteolin-fed mice [17]. The authors concluded that luteolin could attenuate hypercholesterolemia induced by a high-fat diet through the elimination of cholesterol.

## 8. The Effect of Luteolin on the Activity of the NPC1L1 Promoter

It has been reported that some transcriptional factors regulate the expression of NPC1L1, e.g., liver X receptor (LXR) down-regulates NPC1L1 expression [31]. SREBP2-HNF4α and peroxisome proliferator activated receptor α (PPARα) - retinoid X receptor (RXRα) are also thought to be involved in the upregulation of NPC1L1 transcription [32,33]. To examine the effect of luteolin via SREBP2-HNF4α on *NPC1L1* promoter activity, Ogawa et al. made a construct with the *NPC1L1* promoter inserted upstream of a luciferase gene and transiently transfected the vector into HepG2 cells with SREBP2, in the presence or absence of hepatocyte nuclear factor 4α (HNF4α) expression vectors [16]. The luciferase activity of *NPC1L1* promoter was induced by co-transfection of SREBP2, and luteolin reduced this activity. HNF4α alone did not stimulate the transcription of *NPC1L1*, and luteolin did not change the promoter activity. HNF4α stimulated the transcription of *NPC1L1* along with SREBP2, while luteolin reduced this activity. The rate at which luteolin inhibited transcriptional activity when co-transfected with SREBP2-HNF4α was the same as the case with SREBP2 alone. These findings suggest that, via SREBP2 access, luteolin reduced the binding elements in the promoter region on *NPC1L1* promoter activity. Iwayanagi et al., reported that the luciferase activity of NPC1L1 was induced by the co-transfection of PPARα and RXRα [32,33]. Ogawa et al. used these transfection systems to determine whether luteolin affects *NPC1L1* expression through PPARα and RXRα. However, luteolin did not change the PPARα, RXRα, and PPARα-RXRα transcriptional activity [16]. Luteolin decreased *SREBP2* and *HNF4α* mRNA expression, suggesting that this was behind its effect of decreasing the expression of NPC1L1. They also examined the effect of quercetin using the same luciferase assay systems as those of luteolin. However, quercetin did not decrease *NPC1L1* transcriptional activity. 

## 9. The Effect of Luteolin on Hypercholesterolemia, Excluding the Effect on NPC1L1

The effect of luteolin on hypercholesterolemia independent of NPC1L1 is summarized in Table 2. Luteolin suppresses the *LXRα/β* transcriptional activity which, in turn, inhibited *SREBP-1c* expression, lipid accumulation, and *ABCA1* expression [34]. Luteolin suppressed the expression, nuclear translocation, and transcription of SREBP-2 in hepatic cell lines, such as HepG2. The transcription of *HMGCR*, which is the target of SREBP2, was decreased after luteolin treatment [35]. Luteolin suppressed the HNF4α mRNA target genes, and these mechanisms suppressed the acetylation level of histone H3 in the promoter region of certain HNF4α target genes [36]. Luteolin has been reported to inhibit fatty acid synthesis from acetyl-CoA via blocking fatty acid synthase [37]. Several studies have reported that luteolin not only decreased the cholesterol level but also suppressed cardiometabolic alterations, vascular dysfunction, atherosclerosis, and cardiac ischemia/reperfusion injury in models for obesity or hypercholesterolemic via antioxidant and anti-inflammation activity [38,39,40]. Luteolin alleviated alcoholic liver disease induced by chronic and binge ethanol feeding in mice via the inhibition of *Srebp1c*, *Fasn*, *Acc*, and *Scd1* genes expression [41]. Furthermore, luteolin suppressed diabetes-induced impairment via ameliorating detrimental changes in lipid profile, advanced glycation end products, and oxidative stress [42,43,44].

## 10. Conclusions and Future Perspectives

Studies regarding the effects of polyphenols on attenuating hypercholesterolemia through the inhibition of NPC1L1 inhibition of polyphenols are relatively new. It is possible that polyphenols exert multiple effects on NPC1L1. For instance, luteolin reduces high blood cholesterol levels through two different mechanisms. One is the inhibition of NPC1L1 directly, which occurs in a short time frame, such as with ezetimibe [14]. A second mechanism is the inhibition of NPC1L1 expression by luteolin [23]. Luteolin inhibited the expression of NPC1L1 by decreasing SREBP2 and HNF4α expression and inhibiting the access of SREBP2 to binding elements in the promoter region. The present study provides new information into whether the daily intake of luteolin (or vegetables and fruits containing luteolin) may prevent increases in blood cholesterol.

The large-scale clinical trial IMPROVE-IT was published in the New England Journal of Medicine [45]. This was the first study to properly evaluate the clinical effect of adding ezetimibe to statin. They conducted a double-blind, randomized trial involving 18,144 patients who had been hospitalized for an acute coronary syndrome. The ezetimibe–statin therapy combination decreased LDL-C levels and improved cardiovascular outcomes. For LDL-C, its level was significantly lower than the target value of 70 mg/dL, and up to about 54 mg/dL, a stronger effect of suppressing events was shown, and the point that it was shown with safety is the biggest impact. It can also be asserted that, for the first time since the appearance of statins, the effect of novel drugs of suppressing events has been revealed.

It was reported that fat-soluble vitamins are substrates of NPC1L1. For example, the uptake of both cholesterol and α-tocopherol was significantly increased by the overexpression of human NPC1L1, while ezetimibe inhibited their uptake [46]. In an in vivo study, the intestinal permeability of α-tocopherol was inhibited by ezetimibe, although the inhibitory effect was lesser than that of cholesterol. Vitamin E has a variety of homologs and partial homologs that influence intestinal permeability, which is inhibited by ezetimibe. A clinical study reported that the serum concentrations of vitamin E were not significantly altered by a 12-week administration of ezetimibe [47]. These data suggested that, at sufficient levels of vitamin E, its bioavailability depends on the α-tocopherol transfer protein, which is the transport protein from the liver. Furthermore, intestinal vitamin K is an NPC1L1 substrate and inhibited by ezetimibe [48]. Vitamin K facilitates blood coagulation by activating clotting factors such as prothrombin and factors II, VII, IX, and X in the liver [49,50]. Vitamin K is not synthesized in the body and must be obtained by intestinal absorption from the daily diet and products from intestinal microbiota. Warfarin competitively inhibits vitamin K by binding to the C1 subunit of vitamin K epoxide reductase. This suppresses the biosynthesis of the above-mentioned coagulation factors, thereby preventing blood coagulation. In patients treated with warfarin, the intake of vitamin K-rich foods such as fermented soybeans and spinach is restricted because of the potential for attenuating the anticoagulant effect of warfarin. Takada et al. reported that in vivo pharmacological studies have demonstrated that the co-administration of ezetimibe and warfarin caused a reduction in hepatic vitamin K levels and enhanced the pharmacological effect of warfarin [48]. As in the case of vitamin K, excessive NPC1L1 inhibition may cause side effects with medicine, so careful evaluation of this possibility is needed.

On the other hand, regarding blood cholesterol lowering, there are also negative opinions on LDL-C decreasing by drugs. It is also known that there are individual differences in sensitivity to hypercholesterolemia [51]. Regarding the causes of these differences in sensitivity, analysis of their association with genetic polymorphisms is also underway, raising hopes about the further development of this field.

## Figures and Tables

**Figure 1 ijms-20-04939-f001:**
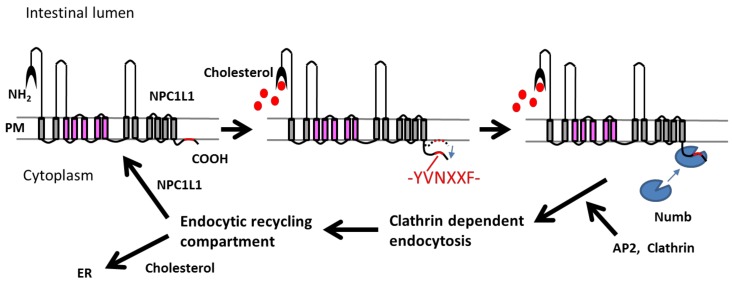
The clathrin-related endocytosis cholesterol transport mechanisms of Niemann–Pick C1-like 1. Modified from Li et al., ([10]).

**Figure 2 ijms-20-04939-f002:**
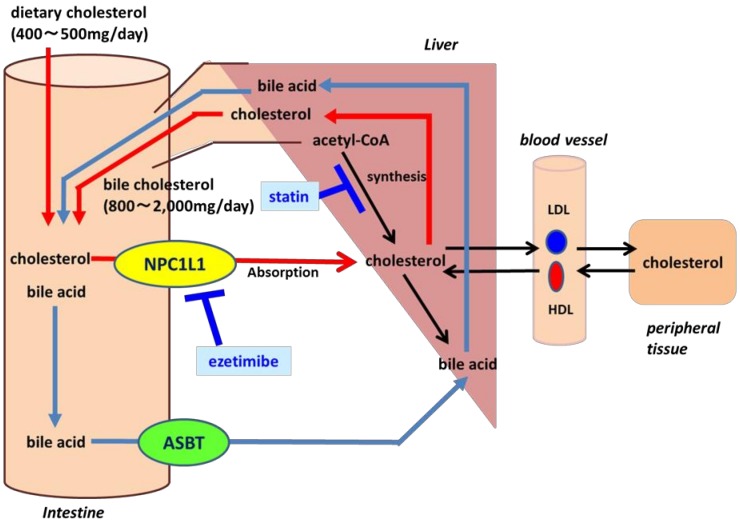
Schematic presentation of blood cholesterol regulation and inhibitory medicines (statin and ezetimibe).

**Figure 3 ijms-20-04939-f003:**
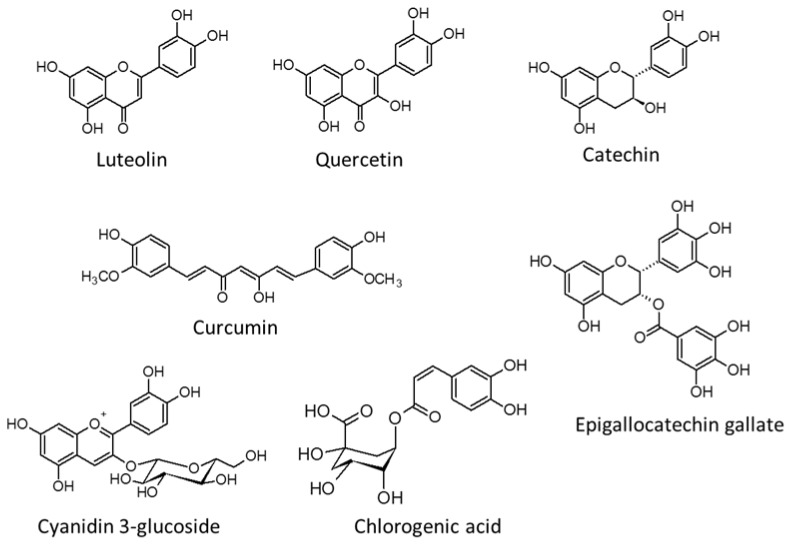
Chemical structures of polyphenols which have been reported to inhibit NPC1L1.

**Table 1 ijms-20-04939-t001:** Flavonoids and flavonoid-rich extracts effect on NPC1L1.

Flavonoids and Flavonoid-Rich Extracts	Model	Results	Reference
Luteolin, quercetin	Caco-2, Wistar rat (male)	Pretreated the Caco-2 cells with 100 µM luteolin for 1 h to inhibit cholesterol uptake. Luteolin (20 mM) and quercetin at 5 mL/kg body weight diet decreased the cholesterol concentration in rat.	[14]
Luteolin	Caco-2	The 100 µM luteolin-treated Caco-2 cells significantly attenuated the NPC1L1 mRNA and protein levels.	[16]
Luteolin	C57BL / 6 mouse (male)	After 8 weeks of treatment, the administration of 250 ppm luteolin could modulate the total and serum non-HDL cholesterol. *Npc1l1* mRNA expressions were decreased in the intestinal mucosa of luteolin-fed animals.	[17]
Curucumin	Caco-2	Pretreated the Caco-2 cells with 25–100µM curcumin for 24 h showing a dose-dependent inhibition of cholesterol uptake. Curcumin decreased the levels of the NPC1L1 protein	[18]
Wild *Lonicera caerulea* berry polyphenol extract (LCBT), cyanidin-3-glucoside, catechin, chlorogenic acid	Caco-2, Wistar rat (male)	Treatment the Caco-2 cells with LCBT and 50 µM cyanidin-3-glucoside, catechin, and chlorogenic acid for 24 h down-regulated *NPC1L1* mRNA expression. LCBP supplementation at 150 and 300mg/kg decreased the TC, TG, and LDL-C levels but increased the HDL-C level. Intestinal NPC1L1 protein expression was reduced in the LCBP treatment group.	[19]
Caffeine-free hawk tea, EGCG	HepG2, Caco2, CRL1601/NPC1L1-EGFP, plasma membrane Sprague–Dawley rat (male)	Hawk tea extract inhibited NPC1L1-mediated free cholesterol uptake, which induced the transcription of low-density lipoprotein receptor downstream the sterol response element binding protein 2 pathway in hepatocytes. EGCG inhibits the endocytosis of NPC1L1 from the plasma membrane to endocytic recycling compartment.	[20]
Cranberry anthocyanin extract (CrA)	Hamster	Plasma total cholesterol and aorta atherosclerotic plaque decreased in a dose-dependent manner with increasing amounts of 1% and 2% CrA added into diets. CrA had no effect on the mRNA levels of intestinal *NPC1L1*.	[21]
Blueberry anthocyanins extract	Hamster	Dietary supplementation of 0.5% and 1.0% blueberry anthocyanins for 6 weeks decreased plasma TC concentration and increased the excretion of fecal neutral and acidic sterols. Blueberry anthocyanins down-regulated the gene expression of *NPC1L1*.	[22]
Aqueous cinnamon extract (CE)	Wistar rat (male) Primary enterocytes	The intestinal *Npc1l1* mRNA levels were lower after treatment for 2 h of 10 μg/mL CE treatment and for 4 h of 100 μg/mL cinnamon extract treatment.	[23]

**Table 2 ijms-20-04939-t002:** The effect of luteolin on hypercholesterolemia, independent of the effect on NPC1L1.

Model	Anti-Hypercholesterolemia Effects	Reference
HepG2, RAW 264.7	Luteolin abrogated the *LXRα/β* transcriptional activity and, subsequently, inhibited *SREBP-1c* expression, lipid accumulation, and *ABCA1* expression.	[34]
HepG2, non-cancer WRL	Luteolin suppressed the expression, nuclear translocation, and transcription of SREBP-2 in the hepatic cell lines. The transcription of *HMGCR* also decreased after luteolin treatment.	[35]
HepG2, Caco-2, C57BL/6 mouse (male)	The activity of the *MTP* gene promoter was suppressed by luteolin. Luteolin decreased the mRNA levels of HNF4α target genes and inhibited apoB-containing lipoprotein secretion. Luteolin suppressed the acetylation level of histone H3 in the promoter region of certain HNF4α target genes. Luteolin treatment of mice for 57 days lowered serum VLDL and LDL cholesterol, and apoB protein levels without accumulating fat in the liver.	[36]
MIA PaCa-2	Luteolin inhibited fatty acid synthesis from acetyl-CoA by blocking fatty acid synthase.	[37]
C57BL/6 mouse(male)	In HFD mice, luteolin counteracted the increase in body, epididymal fat weight, and the associated metabolic alterations. Luteolin restored vascular endothelial NO availability, normalized the media–lumen ratio, decreased ROS and TNF levels, and normalized eNOS, SOD1 and microRNA-214-3p expression.	[38]
C57BL/6 mouse (male)	Luteolin prevents plaque development and lipid accumulation in the abdominal aorta by decreasing macrophage inflammation during atherosclerosis, which is mediated by mechanisms including AMPK-SIRT1 signaling.	[39]
Sprague–Dawley rat(male)	Luteolin protects the hypercholesterolemic heart against ischemia/reperfusion injury due to upregulation of Akt-mediated Nrf2 antioxidative function and inhibition of mitochondrial permeability transition pore.	[40]
C57BL/6 mouse (male), AML-12, L-02	Compared with the EtOH group, the EtOH + luteolin group had reductions in serum ALT, TG, LDL cholesterol, and lipid accumulation in the liver. Luteolin reduced ethanol-induced expression of *Srebp1c, Fasn, Acc*, and *Scd1* genes in the liver. In cultured hepatocytes, luteolin prevented alcohol-induced lipid accumulation and increased the expression of lipogenic genes.	[41]
Wistar rat (male)	Luteolin prevented increase in the levels of TGs, total and LDL cholesterol in a streptozotocin-injected diabetes model.	[42]
Sprague–Dawley rat (male)	Luteolin treatment induced a decrease in serum TG, TC, LDL, MDA, CK, LDH, and myocardial CTGF and a significant increase in HDL, SOD and Akt phosphorylation levels in comparison with a diabetics group.	[43]
Wistar rat (male)	Luteolin-7-glucoside was assessed in vivo in healthy rats for the effects on plasma glucose and lipid profile (total cholesterol, HDL and LDL), as well as liver glycogen content in a diabetes model.	[44]

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
