# Peer review of "The Effect of Polyphenols on Hypercholesterolemia through Inhibiting the Transport and Expression of Niemann–Pick C1-Like 1"

_ijms, 2019, doi:10.3390/ijms20194939_

Round 1

Reviewer 1 Report

In the manuscript ijms-589758 Luteolin Inhibits the Transport and Expression of Niemann-Pick C1-Like 1 in Intestine, the author proposed an interesting review about the activity of Luteolin on NPCL1. The review is well organized and is intended to explain how the target was discovered and the effects of flavonoidsfound. The reader is well helped in explaining this field. So, I think that the manuscript is suitable for publication in IJMS after minor revisions as reported: Line 9: abstract, please remove "," and add "is". It is more suitable. Line 56: please write "molecular target" in spite of "target molecule" Line 92 and line 103: please rewrite the sentences in the third person. Lines 113-114: please remove "flavanones" and use the general term "flavonoids". Quercetin is a flavonol while Luteolin is a flavon. Lines 117-118: please add the doses used in these assays. Line 120: please rewrite the sentences in the third person. Line 125: please rewrite the sentences in the third person. Line 140: please rewrite the sentences in the third person. Line 156: please rewrite the sentences in the third person. Line 198: please rewrite the sentences in the third person. In the review, you refer to C3-OH position of flavonoids, with a distinctive effect for the molecules which contain it. Why other molecules were not presented in your manuscript, in order to highlight the data. Are there other polyphenols which inhibit NPLC1? Please also rewrite the abstract. It is less useful for a review manuscript. Also the title should be modified. In this way, it is useful for an original article.

Author Response

Responses to Reviewer’s comments

Reviewer 1:

In the manuscript ijms-589758 Luteolin Inhibits the Transport and Expression of Niemann-Pick C1-Like 1 in Intestine, the author proposed an interesting review about the activity of Luteolin on NPCL1. The review is well organized and is intended to explain how the target was discovered and the effects of flavonoids found. The reader is well helped in explaining this field. So, I think that the manuscript is suitable for publication in IJMS after minor revisions as reported:

Line 9: abstract, please remove "," and add "is". It is more suitable. (Line 10) Line 56: please write "molecular target" in spite of "target molecule" (Line 57) Line 92 and line 103: please rewrite the sentences in the third person. (Line 99 and 101) Lines 113-114: please remove "flavanones" and use the general term "flavonoids". Quercetin is a flavonol while Luteolin is a flavon. (Lines 173-176) Lines 117-118: please add the doses used in these assays. (Lines 179) Line 120: please rewrite the sentences in the third person. (Line 181) Line 125: please rewrite the sentences in the third person. (Line 186) Line 140: please rewrite the sentences in the third person. (Line 201) Line 156: please rewrite the sentences in the third person. (Line 224) Line 198: please rewrite the sentences in the third person. (Line 139)

Response: Thank you for your advice. I have revised the text accordingly.

In the review, you refer to C3-OH position of flavonoids, with a distinctive effect for the molecules which contain it. Why other molecules were not presented in your manuscript, in order to highlight the data. Are there other polyphenols which inhibit NPLC1?

 Response: There are eight polyphenols that have been reported to inhibit NPC1L1. Luteolin, curcumin, cyanidin-3-glucoside, catechin, chlorogenic acid, and catechin inhibit the NPC1L1 expression, whereas luteolin, quercetin, and EGCG inhibit the transport of NPC1L1. The structure-activity relationship was not demonstrated. Therefore, I have deleted the description regarding the C3-OH position of flavonoids.

Please also rewrite the abstract. It is less useful for a review manuscript. Also the title should be modified. In this way, it is useful for an original article.

Response: Thank you for your advice. I have revised the title and abstract according to your comments.

Reviewer 2 Report

In this manuscript, the author described the discovery and the role of luteolin in the modulation of NPC1L1. The review article is clear and well written, however its contribution to the specific field of research is very weak. A review article should summarize and critically discuss a variety of experimental findings in the context of a specific topic. I really miss the point of proposing a review article when the findings discussed are restricted to only two or three reports. Literature data is very limited, and the natural consequence of this issue consists in a weak and extremely self-referential article.

Author Response

Responses to Reviewer’s comments

Reviewere 2:

In this manuscript, the author described the discovery and the role of luteolin in the modulation of NPC1L1. The review article is clear and well written, however its contribution to the specific field of research is very weak. A review article should summarize and critically discuss a variety of experimental findings in the context of a specific topic. I really miss the point of proposing a review article when the findings discussed are restricted to only two or three reports. Literature data is very limited, and the natural consequence of this issue consists in a weak and extremely self-referential article.

Response: Thank you for your advice. As you mentioned, this paper was a limited review; therefore, I revised the abstract and text in accordance with the comments. Firstly, a paper on regarding the suppression of hypercholesterolemia via NPC1L1 was presented. Luteolin was mentioned in this paper as well as several other published reports. The polyphenols that inhibit NPC1L1 are listed in Figure 3 and described in more detail within section 3.

Reviewer 3 Report

Kobayashi extensively summarized the effect of luteolin on hypercholesterolemia and its mechanism of action (MOA). However, some part of the review is lengthy and/or unclear and therefore the review manuscript requires a revision. The followings are specific sentences/paragraphs that need edits:

Lines 42-43. Re-wording is suggested. Line 73. Why did the author bring up Flotillins? Line 84. Isn't excessive cholesterol stored in peripheral tissues? Lines 113-114. The hypotheses need to be elaborated more. Lines 151-192. This paragraph is very lengthy. Lines 193-212. The author can classify the flavonoids based on different MOAs. (i.e.) State the possible MOAs first, then list the related flavonoids.

In addition, figures should have brief legends. 

Author Response

Responses to Reviewer’s comments

Reviewere 3

Kobayashi extensively summarized the effect of luteolin on hypercholesterolemia and its mechanism of action (MOA). However, some part of the review is lengthy and/or unclear and therefore the review manuscript requires a revision. The followings are specific sentences/paragraphs that need edits:

Lines 42-43. Re-wording is suggested.

 Response: Thank you for the comment. I have revised lines 42–43 accordingly.

Line 73. Why did the author bring up Flotillins?

 Response: Flotillins are proteins that are components of lipid raft. Ge et al. reported that flotillins are essential for the NPC1L1-mediated uptake of cellular cholesterol. I have added this explanation in line 77.

Line 84. Isn't excessive cholesterol stored in peripheral tissues?

 Response: Thank you for your advice. I have added “peripheral tissues” in the sentence on line 89.

Lines 113-114. The hypotheses need to be elaborated more.

 Response: Thank you for your advice. I have revised lines 113–114 (lines 173–176).

Lines 151-192. This paragraph is very lengthy.

 Response: Thank you for your advice. I have divided lines 151–192 (lines 219–262) into two chapters.

Lines 193-212. The author can classify the flavonoids based on different MOAs. (i.e.) State the possible MOAs first, then list the related flavonoids.

 Response: Thank you for your advice. I changed the chapter order. Furthermore, I classified the flavonoids based on different MOAs, and then listed the related polyphenols in chapter 3.

In addition, figures should have brief legends.

 Response: Thank you for your advice. We have added additional legends for all figures.

Round 2

Reviewer 2 Report

The author made a substantial and careful revision of the manuscript. Information reported in this review is still limited and mainly unidirectional, since the majority of the findings on luteolin (which is extensively described if compared to other polyphenols) are mostly provided by the same research group. Despite this limitation, the manuscript appears to be significantly improved.

This manuscript is a resubmission of an earlier submission. The following is a list of the peer review reports and author responses from that submission.